# SUBJECT-DRIVEN VIDEO GENERATION EMERGES FROM EXPERIENCE REPLAYS

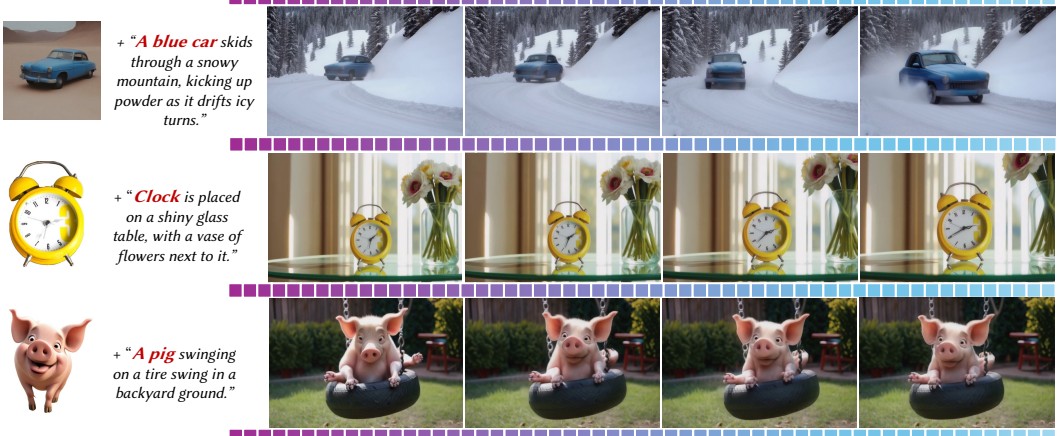

Figure 1: Examples demonstrating high-quality results across various scenarios with different prompt for each reference image.

## ABSTRACT

We aim to enable efficient subject-to-video (S2V) learning, which otherwise requires expensive video-subject-pair datasets that require tens of thousands of GPU hours for training. While utilizing image-paired datasets to train video models could address this challenge, naively training with image pairs results in catastrophic loss of temporal ability due to gradient conflicts. We hypothesize that S2V generation decomposes into two orthogonal objectives of identity learning from images and temporal dynamics from videos. Based on this orthogonality assumption, we design a stochastic task-switching strategy that predominantly samples from image datasets while maintaining minimal video replay for temporal coherence. Our experiments validate this hypothesis by demonstrating that the gradient inner product between tasks converges exponentially to near-zero, confirming emergent orthogonalization without requiring explicit orthogonal projection. This validated orthogonality enables efficient image-dominant training while preventing catastrophic forgetting through proxy experience replay. We employ regularization techniques including random frame selection and token dropping during video replay to ensure efficient temporal learning. Extensive experiments demonstrate our approach achieves superior performance with comparable compute to per-subject tuned methods for single subjects, while providing zero-shot capability and outperforming both per-subject tuned methods and some existing zero-shot approaches.

## 1 INTRODUCTION

Recent advancements in video diffusion models Hong et al. (2022); Yang et al. (2024); Blattmann et al. (2023) have significantly improved controllability by incorporating various conditioning mechanisms, ranging from text-to-video (T2V) synthesis to video customization using key points, edges,

Table 1: Computational comparison on tuning-free and per-subject tuned methods.

| Method | Dataset Size | Base Model (Size) | Train Steps | A100 Hours* | Supporting Domain |
|---|---|---|---|---|---|
| *Tuning-free Methods* | | | | | |
| VACE Jiang et al. (2025) | 53M videos[†] | LTX & Wan (14B) | 200K | 70K-210K[‡] | Face/Object/General |
| Phantom Liu et al. (2025) | 1M pairs[§] | Wan (1.3-14B) & Seed[¶] | 30K | 10K-30K[¶‡] | Face/Object |
| Consis-ID Yuan et al. (2024) | 130K+[**] | CogVideoX (5B) | 1.8K | - | Face |
| *Per-subject Tuned Methods* | | | | | |
| CustomCrafter Wu et al. (2025) | 200 reg. images | VideoCrafter2 (1.4B) | 10K | 200[‡] | Object |
| Still-Moving Chefer et al. (2024) | few ref. images + 40 videos | Lumiere (1.2B) | 500 | - | Face/Object |
| Ours | 200K images + 4K unpaired videos | CogVideoX (5B) | 4K | 288 | Object |

[†]Source pool. [‡]Estimated based on implementation details in paper and GitHub. See supplement for estimation calculation. [§]Phantom-Data Chen et al. (2025b). [¶]Varies by model size. *Total GPU hours. **130K clips, and in terms of pairs, not addressed.

or reference images Meng et al. (2023); Yuan et al. (2024); Atzmon et al. (2024); Wu et al. (2025); Hu & Xu (2023). Among them, *subject-driven video customization*, *i.e.*, subject-to-video (S2V) generation Jiang et al. (2024); Wei et al. (2024a;b); Huang et al. (2025), aims to generate videos that maintain consistent subject identity across different scenes, motions, and contexts. S2V generation has gained significant attention for its wide range of applications, including personalized content creation, marketing, and entertainment. However, early approaches Wei et al. (2024a); Chen et al. (2024); Wu et al. (2025); Ruiz et al. (2023) typically require per-subject optimization, which restricts their applicability due to the additional optimization time.

To eliminate per-subject optimization, recent studies Jiang et al. (2024); Huang et al. (2025); Chen et al. (2025a); Wei et al. (2024b); Liu et al. (2025); Jiang et al. (2025); Hu et al. (2025) have developed zero-shot S2V methods. However, these approaches face critical challenges that require expensive subject-driven video customization datasets and massive computational resources. As shown in Table 1, state-of-the-art tuning-free methods like VACE Jiang et al. (2025) and Phantom Liu et al. (2025) require 70K-210K and 10K-30K A100 hours respectively, training on millions of video-subject pairs. This computational burden stems from training dominantly on video-paired data, which is inherently more expensive than image-based training by orders of magnitude. Recent studies attempt to gather even larger S2V datasets Chen et al. (2025a); Huang et al. (2025); Yuan et al. (2024), but this approach only exacerbates the computational burden without addressing the fundamental problem.

A natural approach to address this challenge is leveraging readily available image customization datasets to train video models, eliminating the need for expensive video-subject pairs. However, naïvely fine-tuning video models on image data results in catastrophic loss of temporal modeling ability, as the model learns to preserve identity but forgets how to generate coherent motion. We hypothesize that this mutual interference can be resolved if subject-driven video generation naturally decomposes into two orthogonal objectives in the gradient space, namely identity learning from images ($\nabla_\theta \mathcal{L}_{\text{img}}$) and temporal dynamics from videos ($\nabla_\theta \mathcal{L}_{\text{vid}}$). If these gradients are orthogonal, they can be optimized independently without interference, enabling efficient training predominantly on cheaper image data.

Based on this orthogonality hypothesis, we formulate subject-driven video generation as a pseudo-continual learning problem and design a proxy experience replay mechanism. Our approach implements stochastic task switching that alternates between identity learning from image customization datasets and temporal preservation through minimal video replay. Since the original pretraining data is unavailable, we use proxy video samples that approximate the pretraining distribution to maintain temporal capabilities. This stochastic interleaving strategy creates a weighted gradient $\mathbb{E}[\mathbf{g}] = (1-p)\nabla_\theta \mathcal{L}_{\text{img}} + p\nabla_\theta \mathcal{L}_{\text{vid}}$, where $p$ is the replay ratio. We further introduce regularization techniques including random frame selection and image-token dropping during video replay phases to ensure effective temporal learning even with minimal video samples.

Our experiments validate the orthogonality hypothesis through empirical observation of gradient dynamics. We observe that the gradient inner product $\langle \nabla_\theta \mathcal{L}_{\text{img}}, \nabla_\theta \mathcal{L}_{\text{vid}} \rangle$ indeed converges to near-zero in an exponential-decay manner, confirming emergent orthogonalization without requiring explicit orthogonal projections. This phenomenon validates our initial hypothesis and explains why proxy replay is remarkably effective. Rather than forcing gradients to be orthogonal through complex projections like PCGrad Yu et al. (2020), the proxy replay dynamics naturally evolve the optimization to find mutually compatible directions for both tasks.

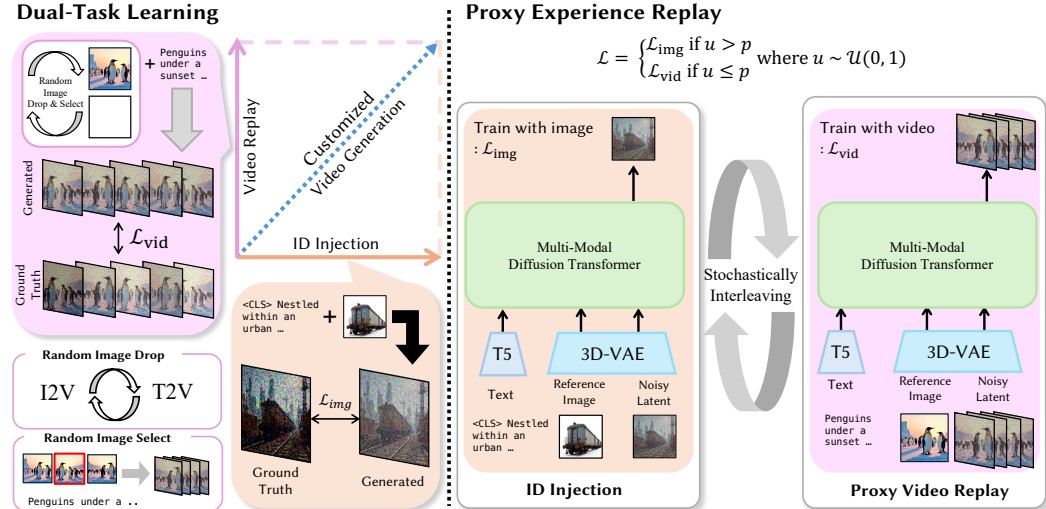

Figure 2: **Overview of our framework.** We interpret the subject-driven video customization (S2V) as dual-task learning with proxy-experience replay in terms of continual learning of the two domains with temporal-awareness preservation and ID injection (Left). To optimize the two objectives, we utilize a stochastically interleaving strategy, randomly switching between the two training objectives (Right).

The validated orthogonality enables computational cost reduction. Our approach achieves zero-shot capability with 288 A100 hours of training (Table 1), comparable to CustomCrafter Wu et al. (2025) (200 A100 hours) which requires per-subject optimization for each new identity. By leveraging predominantly cheap image-based training (80% of iterations) while maintaining minimal unpaired video replay (20%), we reduce computational cost by 92.7%. The emergent gradient orthogonalization ensures both tasks are optimized efficiently without interference, making high-quality video customization both accessible and practical.

The key contributions of our work are as follows:

- We propose the hypothesis that subject-driven video generation decomposes into orthogonal tasks and design a proxy experience replay method based on this assumption.

- We experimentally validate that proxy replay induces emergent gradient orthogonalization, with gradient conflicts naturally resolving to near-zero correlation without explicit projection.

- Our approach achieves superior performance with dramatically reduced computational requirements, outperforming per-subject methods and matching some zero-shot baselines while requiring less compute.

## 2 RELATED WORK

### 2.1 SUBJECT-DRIVEN IMAGE GENERATION

Recent diffusion models Esser et al. (2024); Labs (2024); Chen et al. (2023) have expanded text-to-image synthesis capabilities Meng et al. (2022); Zhang et al. (2023), with a key challenge being injecting novel subjects while maintaining identity across diverse prompts. Early methods like ControlNet Zhang et al. (2023) and T2I-Adapter Fu et al. (2023) used spatially aligned conditioning but struggled with pose variations. IP-Adapter Ye et al. (2023) and SSR-Encoder Zhang et al. (2024) addressed this through cross-attention mechanisms for robust feature integration. Dream-Booth Ruiz et al. (2023) and Textual Inversion Gal et al. (2022) introduced specialized embeddings, while recent works explore tuning-free Ding et al. (2024); Zeng et al. (2024); Tan et al. (2024), multi-subject Kumari et al. (2023); Liu et al. (2023), and subject-agnostic Chan et al. (2024) ap-

proaches. These image-level methods provide foundations for video generation, where temporal coherence adds complexity.

## 2.2 SUBJECT-DRIVEN VIDEO GENERATION

Traditional video generation requires expensive training on large datasets jovianzm (2025); Bain et al. (2021); pan (2022). Recent methods divide into two approaches: (1) test-time optimization methods like DreamVideo Wei et al. (2024a), MotionBooth Wu et al. (2024), Still-Moving Chefer et al. (2024), and CustomCrafter Wu et al. (2025) that separate appearance and motion modules; (2) zero-shot solutions including Consis-ID Yuan et al. (2024), Concept-Master Huang et al. (2025), VideoBooth Jiang et al. (2024), and MagicMirror Zhang et al. (2025) that avoid fine-tuning. Methods like Phantom Liu et al. (2025) and VACE Jiang et al. (2025) achieve strong zero-shot performance but still require video-paired datasets Chen et al. (2025b); Yuan et al. (2025).

## 2.3 EXPERIENCE REPLAY AND CONTINUAL LEARNING

Continual learning prevents catastrophic forgetting French (1999) through experience replay Mnih et al. (2015), which stores and replays past experiences. Variants include exact replay Lopez-Paz & Ranzato (2017), generative replay Shin et al. (2017), and pseudo-rehearsal using proxy samples Robins (1995). GEM Lopez-Paz & Ranzato (2017) constrains gradient updates to preserve previous task performance, while A-GEM Chaudhry et al. (2019) approximates these constraints efficiently. We apply these principles to video generation, where gradient conflicts between identity and temporal objectives necessitate replay mechanisms, even in the absence of sequential task presentation.

Unlike methods that require extensive video training Chen et al. (2025a); Huang et al. (2025); Jiang et al. (2024), we utilize proxy experience replay to avoid the need for large-scale annotated datasets. Similar to pseudo-rehearsal Robins (1995), we use proxy video samples to maintain temporal capabilities while learning identity from images. We formulate S2V as pseudo-continual learning, strategically interleaving image and video samples to prevent forgetting while achieving computational efficiency through predominant use of cheaper image data.

## 3 METHOD

### 3.1 PRELIMINARIES

Our framework builds upon the Multi-Modal Diffusion Transformer (MM-DiT) Peebles & Xie (2023), employed in architectures such as FLUX.1 Labs (2024), Stable Diffusion 3 Esser et al. (2024), CogVideo Hong et al. (2022); Yang et al. (2024) and Wan 2.1 WanTeam et al. (2025). DiT adopts a Transformer-based denoising network that iteratively refines noisy tokens through multi-modal attention.

At each denoising step, DiT processes noisy visual tokens $\mathbf{X} \in \mathbb{R}^{N \times d}$ and text tokens $\mathbf{C}_T \in \mathbb{R}^{M \times d}$, sharing embedding dimension $d$. Each DiT block consists of Layer Normalization (LN) followed by Multi-Modal Attention (MMA). Spatial positions are encoded using Rotary Position Embedding (RoPE) Su et al. (2023) as $\mathbf{X}_{i,j} \to \mathbf{X}_{i,j} \cdot \mathbf{R}(i,j)$, where $\mathbf{R}(i,j)$ is a rotation matrix. MMA computes attention as

$$\text{MMA}\big([\mathbf{X}; \mathbf{C}_T]\big) = \text{softmax}\left(\frac{\mathbf{Q}\mathbf{K}^\top}{\sqrt{d}}\right)\mathbf{V}, \tag{1}$$

where $[\mathbf{X}; \mathbf{C}_T]$ denotes concatenated tokens. The quadratic complexity $O(n^2)$ makes video-paired training computationally expensive.

### 3.2 PROBLEM FACTORIZATION BASED ON ORTHOGONALITY HYPOTHESIS

To adapt the pretrained T2V model to subject-driven video customization without expensive video-subject pairs, we hypothesize that the learning problem can be factorized into two orthogonal objectives: identity injection to learn subject features from S2I dataset Tan et al. (2024) comprising image pairs of the same subject, and temporal awareness preservation to maintain motion dynamics using unpaired video dataset as proxy experiences.

Our core hypothesis is that these two objectives are orthogonal in the gradient space, meaning $\langle \nabla_\theta \mathcal{L}_{\text{img}}, \nabla_\theta \mathcal{L}_{\text{vid}} \rangle \approx 0$. If true, this orthogonality would enable efficient optimization without mutual interference, allowing us to train predominantly on cheaper image data while maintaining temporal coherence through minimal video replay.

### 3.3 Proxy Experience Replay Based on Orthogonality Hypothesis

#### 3.3.1 Orthogonality Hypothesis and Method Design

We hypothesize that identity learning and temporal dynamics constitute orthogonal tasks. Let $\mathbf{g}_1(t) = \nabla_\theta \mathcal{L}_{\text{img}}(\theta_t)$ and $\mathbf{g}_2(t) = \nabla_\theta \mathcal{L}_{\text{vid}}(\theta_t)$ be the gradients for identity and temporal objectives respectively. Our hypothesis states that these gradients should be approximately orthogonal, with gradient conflict

$$\phi(t) = \cos \angle(\mathbf{g}_1(t), \mathbf{g}_2(t)) = \frac{\mathbf{g}_1(t)^\top \mathbf{g}_2(t)}{\|\mathbf{g}_1(t)\|\|\mathbf{g}_2(t)\|} \approx 0 \tag{2}$$

Based on this hypothesis, we design a proxy experience replay mechanism with stochastic task switching. We sample with probability $p$ from video data and $(1-p)$ from image data, creating the weighted update $\theta_{t+1} = \theta_t - \eta[(1-p)\mathbf{g}_1 + p\mathbf{g}_2]$. If our orthogonality hypothesis holds, this simple weighted averaging should enable efficient optimization without gradient interference.

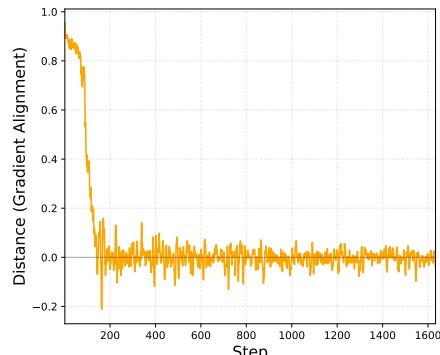

Figure 3: **Gradient Conflict and Alignment.** Our Proxy Replay strategy exponentially converges to zero.

#### 3.3.2 Experimental Validation of Orthogonalization

Our experiments validate the orthogonality hypothesis. As shown in Figure 3, we observe that gradient conflict $\phi(t)$ indeed converges exponentially to zero from an initial negative value ($\phi(0) < 0$), confirming emergent orthogonalization.

**Theorem 1.** Under proxy replay, the gradient conflict $\phi(t)$ converges to zero, validating our orthogonality hypothesis.

*Proof sketch.* The weighted update creates loss dynamics

$$\mathcal{L}_1(\theta_{t+1}) = \mathcal{L}_1(\theta_t) - \eta[(1-p)\|\mathbf{g}_1\|^2 + p\mathbf{g}_1^\top \mathbf{g}_2] + O(\eta^2) \tag{3}$$

where the cross-term $\mathbf{g}_1^\top \mathbf{g}_2$ appears with opposite signs in each loss update, creating repulsion from conflict regions. The gradient alignment $A(t) = \mathbf{g}_1^\top \mathbf{g}_2$ evolves as $A(t) \approx A(0)e^{-\eta\lambda t}$, yielding exponential decay to orthogonality. This validates our initial hypothesis that two tasks can be orthogonal. See supplement for detailed proof.

### 3.4 Task 1: Identity Injection

For identity injection, we adopt the S2I approach Tan et al. (2024). Given source image $I_{\text{input}} \in \mathbb{R}^{H \times W \times 3}$ and target $I_{\text{output}} \in \mathbb{R}^{H \times W \times 3}$ with prompt $P$, we encode using $\mathbf{X}_{\text{in}} = \text{VAE}(I_{\text{input}})$, $\mathbf{X}_{\text{out}} = \text{VAE}(I_{\text{output}})$, and $\mathbf{C}_T = \text{T5}(P)$. We apply LoRA Hu et al. (2022) for parameter-efficient fine-tuning, updating LN layers only for $\mathbf{X}_{\text{in}}$ while keeping them frozen for $\mathbf{X}_{\text{out}}$ and $\mathbf{C}_T$. This selective updating ensures identity learning remains orthogonal to temporal and textual representations, aligning with our hypothesis that identity injection and temporal dynamics are independent objectives.

Additionally, We introduce `<CLS>` token prepended to prompts (*e.g.*, "An `<CLS>` armchair in the living room") to explicitly signal identity mapping. This token acts as an anchor for identity features while maintaining orthogonality with temporal learning. See supplement for ablation on the `<CLS>` token.

Table 2: Quantitative comparison with other methods on VBench.

| Training Method | Used Data | Motion Smoothness | Dynamic Degree | CLIP-T | CLIP-I | DINO-I |
|---|---|---|---|---|---|---|
| VideoBooth | Custom T2V | 96.95 | 51.67 | 29.59 | 66.06 | 34.54 |
| OmniControl+I2V | Custom T2I | 98.21 | 51.67 | 31.89 | 72.58 | 54.16 |
| BLIP+I2V | Custom T2I | 97.53 | 49.17 | 28.19 | **79.29** | 56.58 |
| IP-Adapter+I2V | Custom T2I | 97.21 | 55.83 | 26.97 | 73.86 | 45.18 |
| Ours | Custom T2I | **98.72** | **60.19** | **32.24** | 73.70 | **59.29** |

### 3.5 Task 2: Temporal Awareness Preservation

While S2I training injects identity, it causes loss of temporal awareness. Leveraging our orthogonality hypothesis, we introduce I2V fine-tuning using unpaired videos as proxy experiences to restore temporal dynamics without disrupting the learned identity features.

**I2V vs. T2V Fine-Tuning.** We choose I2V alignment over T2V as it better matches our modalities (image input, video output), maintaining the orthogonal decomposition between identity and motion. T2V training would entangle text-based identity with temporal dynamics, violating our orthogonality assumption. I2V preserves this separation by using visual identity inputs while learning temporal coherence.

**Mitigating First-Frame Overreliance.** Naive I2V training causes copy-and-paste artifacts where the model simply replicates the first frame. To maintain orthogonality between static identity and dynamic motion, we employ random-frame selection by choosing reference frame $i \sim \text{Uniform}(1, T)$ and image-token dropping with probability $p_{\text{drop}}$. These techniques force the model to synthesize motion from partial information rather than copying, preserving the independence between identity learning ($\mathcal{L}_{\text{img}}$) and temporal dynamics ($\mathcal{L}_{\text{vid}}$).

### 3.6 Training with Proxy Experience Replay

Building on our validated orthogonality hypothesis, we implement training through stochastic interleaving. At each iteration, we sample $u \sim \mathcal{U}(0, 1)$ and select the training objective based on replay probability $p = 0.1$, using predominantly cheaper image data (90% of iterations) while maintaining temporal coherence through minimal video replay.

**Objectives.** For S2I pair $(I^{(1)}, I^{(2)})$, we optimize $\mathcal{L}_{\text{img}}(I^{(1)}, I^{(2)})$, and for video $(T, V)$, we optimize $\mathcal{L}_{\text{vid}}(T, V)$, both following v-prediction Yang et al. (2024):

$$\mathcal{L}_{\text{total}} = \begin{cases} \mathcal{L}_{\text{vid}}(T, V), & \text{with probability } p \\ \mathcal{L}_{\text{img}}(I^{(1)}, I^{(2)}), & \text{with probability } 1 - p \end{cases} \quad (4)$$

The experimentally validated orthogonalization ensures both objectives are preserved without interference despite the imbalanced sampling.

**Proxy Replay vs. S2I+I2V.** An alternative S2I+I2V pipeline (Figure 4) sequentially trains identity then motion, but fails when subjects are small or occluded. Our proxy replay jointly learns both objectives through interleaved training, enabling robust identity maintenance across scales while preserving temporal coherence.

**Computational Efficiency.** With video processing cost $C_{\text{vid}} \approx 169 \times C_{\text{img}}$ (13 latent frames, $169 = 13^2$), our approach yields $\mathbb{E}[C] = 0.8 C_{\text{img}} + 0.2 C_{\text{vid}} = 13.72 C_{\text{img}}$, achieving 92.7% reduction. The validated orthogonality enables this efficiency without performance degradation.

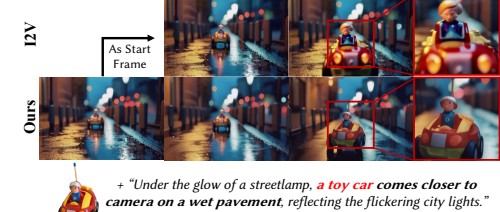

Figure 4: **Limitation of S2I+I2V method.** With subject presented small in first frame, I2V fails to generate consistent results as it cannot interpret low-resolution subjects.

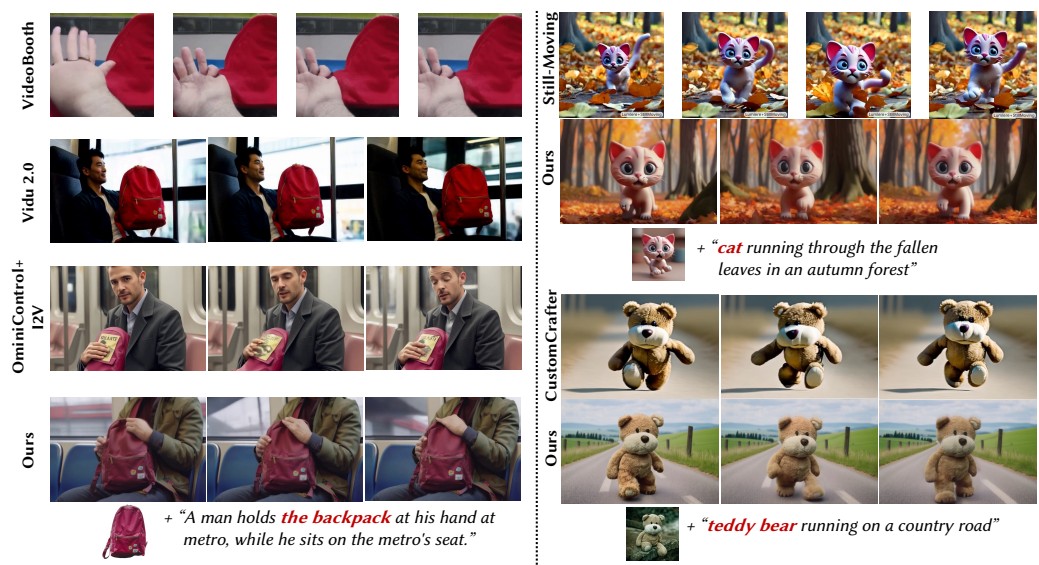

Figure 5: **Qualitative comparison with tuning-free methods (left) and per-subject optimized methods (right).**

## 4 EXPERIMENTS

### 4.1 SETUP

**Implementation Details.** Our method is built upon the CogVideoX-5B, which employs a 3D Multi-modal Diffusion Transformer (MM-DiT) architecture utilizing a v-prediction objective Yang et al. (2024) with DDIM scheduler following Rombach et al. (2021). Following the standard CogVideoX training pipeline, we adopt the AdamW optimizer with a learning rate of $5 \times 10^{-5}$ and a cosine with restart learning rate scheduler. Training was performed for 4,000 steps, requiring approximately 288 GPU hours on NVIDIA A100 GPUs. We utilize a batch size of 256 for image data and 32 for video data during joint fine-tuning. To facilitate efficient training, we use mixed-precision BF16 training. Our method employs LoRA (Low-Rank Adaptation) with rank 128 and dropout of 0.1 for efficient fine-tuning. We set the probability parameter $p = 0.1$ for stochastic switching, ensuring balanced optimization between identity injection and temporal modeling. We utilize OmniControl's Subject200K dataset for image customization, comprising 200K subject-specific image pairs covering various poses, styles, and contexts. Additionally, for video fine-tuning, we leverage 2.5% of the Pexels 400K dataset by randomly selecting approximately 10,000 unpaired videos. Note that for ablation study, we used CogVideoX and provide details in supplement.

**Baseline.** For baseline comparison, we utilize video S2V model VideoBooth Jiang et al. (2024), and state-of-the-art image S2I models Tan et al. (2024); Li et al. (2023); Ye et al. (2023) along with image-to-video (I2V) model. We utilize OmniControl Tan et al. (2024), and BLIP-Diffusion Li et al. (2023) and IP-Adapter Ye et al. (2023). Each baseline initially performs image customization independently using its original setup, after which we apply the CogVideoX-5B I2V model to get video from the images. These methods represent state-of-the-art in subject-driven video customization, allowing us to comprehensively evaluate the effectiveness of our proposed approach under zero-shot conditions. Also for methods with per-subject tuning, we compare with state-of-the-art S2V models that requires per-subject tuning, Still-Moving Chefer et al. (2024) and CustomCrafter Wu et al. (2025). Since the code is neither public Chefer et al. (2024) or requires extensive finetuning for each sample Wu et al. (2025), we compare with samples inside their paper and supplement.

**Evaluation.** We gather 30 reference images from state-of-the-art image customization papers Avrahami et al. (2023); Chefer et al. (2024) along with the traditional DreamBooth dataset Ruiz et al. (2023). We utilize GPT to generate four prompts for each image and evaluate using VBench Huang et al. (2024).

Table 3: Ablation result on training strategy of alternating optimization with image-only and two-stage training approaches.

| Training Method | Motion Smoothness | Dynamic Degree | CLIP-T | CLIP-I | DINO-I |
|---|---|---|---|---|---|
| Image-only | 99.60 | 0.84 | 32.67 | 71.15 | 43.19 |
| Two-stage | 96.04 | 81.51 | 28.96 | 84.73 | 76.13 |
| Ours | 98.72 | 60.19 | 32.24 | 73.70 | 59.29 |

Additionally, we assess temporal modeling performance using 300 videos sampled from the Pexels dataset, ensuring no overlap with videos used during training. Following FloVD Jin et al. (2025), we classify these videos into three groups based on optical flow magnitude (small: $\leq 25$, medium: $25 \sim 50$, large: $\geq 50$) for detailed analysis. We additionally demonstrate the preprocessing details to evaluate temporal modeling performance in the supplement.

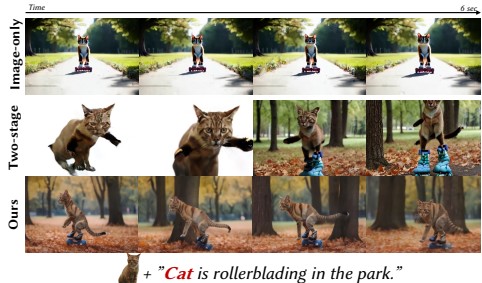

+ "**Cat** is rollerblading in the park."

Figure 6: Qualitative result on ablation study of our component in temporal awareness preservance.

### 4.2 COMPARISON WITH TUNING-FREE METHODS

**Quantitative Analysis.** Table 2 shows our method outperforming baselines in zero-shot video customization across motion smoothness (98.72), dynamic degree (60.19), text alignment (CLIP-T, 32.24), and identity consistency (DINO-I, 59.29). Notably, we achieve significant gains over Video-Booth in identity preservation (DINO-I: +24.75) and prompt fidelity (CLIP-T: +2.65). Compared to OmniControl + I2V, our method enhances dynamic degree (+8.52) and temporal coherence (DINO-I: +5.13). While BLIP-Diffusion achieves a higher CLIP-I (79.29), our approach offers a more balanced performance, excelling in diverse motion and subject fidelity, underscoring its effectiveness and generalization.

**Qualitative Analysis.** As illustrated in Figure 5, our method demonstrates superior detail retention, ID consistency, and natural temporal transitions compared to OmniControl + I2V, Vidu 2.0, and VideoBooth. For the backpack example (Fig. 5, left), our model faithfully reproduces intricate details. In contrast, Vidu 2.0 preserves ID but shows erratic movement; OmniControl + I2V captures the subject reasonably but with occasional artifacts; and VideoBooth yields the weakest ID fidelity. Overall, our approach better preserves subject identity and coherent motion across examples, aligning with quantitative improvements.

Table 4: Temporal Evaluation following FloVD Jin et al. (2025), assessing whether motion dynamics improve compared to image-only or two-stage training. Small - Medium - Large - with each number representing FVD. † denotes Pexels jovianzm (2025)-finetuned version of CogVideoX Yang et al. (2024).

| Method | Small↓ | Medium↓ | Large↓ |
|---|---|---|---|
| CogVideoX-T2V† | 597.54 | 594.26 | 573.86 |
| Image-only | 641.92 | 636.42 | 680.34 |
| Two-stage | 801.97 | 872.30 | 824.03 |
| Ours | **512.30** | **511.66** | **550.14** |

### 4.3 COMPARISON WITH PER-SUBJECT OPTIMIZED METHODS

We show qualitative comparisons with Still-Moving Chefer et al. (2024) and CustomCrafter Wu et al. (2025) using their official samples in the Figure 5 right. Against Still-Moving Chefer et al. (2024), Our method exhibits superior identity preservation. The "pink cat" (row 1) has more consistent color and detailed whiskers; the "pig" (row 2) shows more faithful eye shape and coloration, indicating higher subject fidelity. Also compared to CustomCrafter Wu et al. (2025) (rows 3–4), Our results are more identity-faithful, whereas CustomCrafter often displays distorted details.

## 4.4 ABLATION STUDY

**Training Strategy.** We compared *image-only*, *two-stage*, and our *proxy experience replay* training strategies. Image-only training produced superficially smooth (99.60) but static videos (0.84 dynamic degree), with poor FVD (Tables 3, 4, Fig. 6). Two-stage training improved dynamics (81.51) and ID similarity (CLIP-I: 84.73) but introduced severe artifacts and S2V forgetting, leading to high FVD. Our *alternating* strategy (Ours) excelled, balancing motion smoothness (98.72) and dynamic degree (60.19) without the drawbacks of the other methods. Its lower FVD scores confirm superior motion realism and temporal consistency, on par with CogVideoX Yang et al. (2024) (see supplement).

**Random Initial Frame Selection & Dropping.** We also examined the effect of *random frame selection* and *image-token dropping* on I2V fine-tuning. Without either technique, the model often shows superficially strong motion smoothness but drastically reduced dynamic degree, indicating a near-static outcome where the subject barely moves. Qualitative observations (Figure 7) reveal that the first reference image dominates subsequent frames, inflating identity metrics like CLIP-I and DINO-I while eliminating motion.

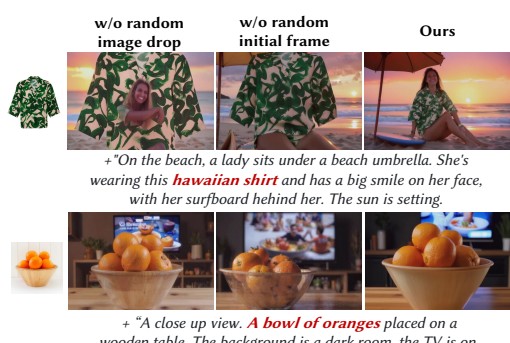

Figure 7: Effect of random initial frame and image token dropping.

By enabling only *random frame selection*, we recover some degree of temporal variability yet introduce artifacts in scenes such as shirts or oranges, thereby lowering image alignment scores. By contrast, the combination of frame selection and token dropping strikes an effective balance. We can observe that artifacts are greatly reduced with the reference being more naturally blended. This confirms the importance of mitigating first-frame over-reliance and excessive conditioning on a single reference image for smoother, more natural video generation.

## 4.5 LIMITATIONS

Our primary finetuning dataset, the Subject-200K dataset Tan et al. (2024), primarily comprises general objects and contains few human faces. Although our method supports video customization for arbitrary inputs, training on such a general dataset may hinder its effectiveness for human-specific personalization. When evaluated on human facial data as in Figure 8, our approach generates recognizable facial structures but often introduces blurring artifacts, thereby failing to preserve nuanced identity features. We believe incorporating an identity-focused dataset would mitigate this is-

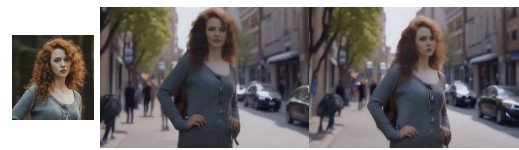

Figure 8: **Limitation of our work.** Identity preservation with human faces shows blurry artifacts since Subject200K Tan et al. (2024), which our model trained lacks human identities.

sue and identify this as a key area for future investigation. Additionally, the theoretical analysis makes simplifying assumptions that might not hold for deep networks and requires further analysis on this in future work.

## 5 CONCLUSION

We presented a zero-shot subject-driven video generation method based on the hypothesis that identity learning and temporal dynamics are orthogonal tasks. Our approach bypasses expensive S2V datasets through proxy experience replay with stochastic task switching between S2I image pairs and unpaired videos. Experiments validate our orthogonality hypothesis, showing gradient conflicts converge to near-zero, enabling computational reduction with image-dominant training, outperforming per-subject tuned models and some tuning-free models.

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

# A   APPENDIX

---

**Algorithm 1: Proxy Experience Replay for Subject-Driven Video Generation**

---

**Input**  : Pretrained MM-DiT model $f_\theta$;
    Image dataset $\mathcal{D}_{\text{img}}$ (pairs $(I^{\text{src}}, I^{\text{tgt}}, \mathcal{P})$ for identity);
    Proxy video dataset $\mathcal{D}_{\text{vid}}$ (videos $(V, \mathcal{T})$ as proxy experiences);
    Replay probability $p \in [0, 1]$ controlling task interleaving;
    `RandomFrameSelect()` and `TokenDrop()` regularizations.

**Output:** Fine-tuned model $f_\theta$ achieving emergent gradient orthogonalization

**Problem Decomposition (Sec. 3.2):**

   • *Task 1 - Identity Learning*: Learn from $\mathcal{D}_{\text{img}}$ for subject features

   • *Task 2 - Temporal Preservation*: Maintain dynamics via proxy replay from $\mathcal{D}_{\text{vid}}$

**Proxy Experience Replay (Sec. 3.3):**

```
/* Initialize gradient tracking for orthogonalization monitoring    */
```
$\phi_0 \leftarrow \cos(\nabla_\theta \mathcal{L}_{\text{img}}, \nabla_\theta \mathcal{L}_{\text{vid}})$                              `// Initial conflict`

**foreach** *iteration* $t \leftarrow 1, \ldots, T_{max}$ **do**

   Sample $u \sim \mathcal{U}(0, 1)$ for stochastic task selection

   **if** $u < p$ **then**

```
        /* Proxy Replay:  Sample proxy experiences to preserve temporal
            dynamics                                                    */
```
        Fetch batch $\{(V_i, \mathcal{T}_i)\}$ from $\mathcal{D}_{\text{vid}}$                              `// Proxy samples`
        $\mathbf{f}_{\text{ref}} \leftarrow$ `RandomFrameSelect`$(V_i)$                              `// Prevent overfitting`
        $\mathbf{X}_{\text{ref}} \leftarrow$ `TokenDrop`$(\text{VAE}(\mathbf{f}_{\text{ref}}), p_{\text{drop}})$
        Compute $\mathcal{L}_{\text{vid}}(V_i, \mathbf{X}_{\text{ref}}, \mathcal{T}_i)$ via v-prediction
        $\theta \leftarrow \theta - \eta \mathbf{g}_2$

   **else**

```
        /* Identity Learning:  Sample image pairs for subject features */
```
        Fetch batch $\{(I_i^{\text{src}}, I_i^{\text{tgt}}, \mathcal{P}_i)\}$ from $\mathcal{D}_{\text{img}}$
        Prepend `<CLS>` token to $\mathcal{P}_i$ for identity signaling
        Apply LoRA updates for parameter-efficient learning
        Compute $\mathcal{L}_{\text{img}}(I_i^{\text{src}}, I_i^{\text{tgt}}, \mathcal{P}_i)$ via v-prediction
        $\mathbf{g}_1 \leftarrow \nabla_\theta \mathcal{L}_{\text{img}}$                              `// Identity gradient`
        $\theta \leftarrow \theta - \eta \mathbf{g}_1$

   **end**

```
   /* Monitor emergent orthogonalization                              */
```
   **if** $t \mod 100 = 0$ **then**

      $\phi_t \leftarrow \cos(\mathbf{g}_1, \mathbf{g}_2)$                              `// Track convergence to zero`

   **end**

**end**

```
/* Expected outcome:  φ_t → 0 (orthogonality) after ~1K iterations    */
```
**return** $f_\theta$ *with naturally orthogonalized task gradients*

---

In this section, we provide additional details on our zero-shot subject-driven video (S2V) framework, including full pseudocode for the proposed method and extended ablation results that further validate our design choices.

## A.1   METHOD DETAILS

Our subject-driven video (S2V) framework, as outlined in Algorithm 1, begins by initializing a pre-trained multi-modal diffusion transformer $f_\theta$ (for example, CogVideoX Yang et al. (2024)), which already possesses general text–visual alignment and motion priors. Two datasets are made available: the first is an image-based S2I dataset $\mathcal{D}_{\text{S2I}}$ containing image pairs $(I^{(1)}, I^{(2)})$ depicting the same subject under varying poses or contexts, and the second is an unlabeled video dataset $\mathcal{D}_{\text{vid}}$ consisting of text–video pairs $(T, V)$. Our key insight is to *factorize* video customization into two tasks—identity injection and temporal modeling—and then interleave them stochastically during training.

At each training iteration, a uniform random variable $u$ is drawn from $\mathcal{U}(0, 1)$. If $u < p$, we sample a mini-batch of unlabeled video $(T_i, V_i)$ and perform *temporal-phase* optimization (lines 9–14 in Algorithm 1), referred to as I2V fine-tuning. This phase leverages the random frame selection and

image-token dropping steps (`Random Frame Select` and `Random Image Drop`) to discourage the model from fixating on a single reference frame. By computing the video reconstruction loss $\mathcal{L}_{\text{vid}}(T_i, V_i)$ in a v-prediction manner, we update $f_\theta$ to maintain or recover realistic motion characteristics. Alternatively, if $u \geq p$, we focus on *identity-phase* optimization (lines 16–19), sampling a mini-batch of image pairs $(I_i^{(1)}, I_i^{(2)})$ from $\mathcal{D}_{\text{S2I}}$ and minimizing the identity injection loss $\mathcal{L}_{\text{img}}(I_i^{(1)}, I_i^{(2)})$. Crucially, we only tune the LoRA-based parameters dealing with the subject-specific tokens $\mathbf{X}_{\text{in}}$ in this phase, preserving the model's capacity to handle text tokens $\mathbf{C}_T$ and output frames $\mathbf{X}_{\text{out}}$.

By stochastically switching between these two objectives, the model balances subject fidelity and temporal consistency throughout training, avoiding the pitfalls of purely sequential or single-focus approaches. After $T_{\text{max}}$ iterations, $f_\theta$ emerges as a zero-shot S2V model capable of generating videos that simultaneously preserve the subject's identity and exhibit coherent motion—even though no large-scale annotated S2V dataset was required.

### A.2   Theoretical Analysis of Emergent Gradient Orthogonalization

#### A.2.1   Problem Setup and Definitions

Consider two loss functions $\mathcal{L}_1 : \mathbb{R}^n \to \mathbb{R}$ (identity loss) and $\mathcal{L}_2 : \mathbb{R}^n \to \mathbb{R}$ (temporal loss) with gradients $\mathbf{g}_1(\theta) = \nabla_\theta \mathcal{L}_1(\theta)$ and $\mathbf{g}_2(\theta) = \nabla_\theta \mathcal{L}_2(\theta)$.

**Definition A.1 (Gradient Conflict).** The gradient conflict at parameters $\theta$ is:

$$\phi(\theta) = \cos \angle(\mathbf{g}_1(\theta), \mathbf{g}_2(\theta)) = \frac{\langle \mathbf{g}_1(\theta), \mathbf{g}_2(\theta) \rangle}{\|\mathbf{g}_1(\theta)\| \|\mathbf{g}_2(\theta)\|} \tag{5}$$

**Definition A.2 (Proxy Experience Replay).** The stochastic gradient update with replay probability $p \in (0, 1)$ is:

$$\theta_{t+1} = \theta_t - \eta \mathbf{g}_t, \quad \text{where} \quad \mathbf{g}_t = \begin{cases} \mathbf{g}_2(\theta_t) & \text{w.p. } p \\ \mathbf{g}_1(\theta_t) & \text{w.p. } 1 - p \end{cases} \tag{6}$$

#### A.2.2   Main Theorem

**Theorem A.1 (Convergence to Orthogonality).** Under the following assumptions:

1. Both $\mathcal{L}_1$ and $\mathcal{L}_2$ are $L$-smooth: $\|\nabla^2 \mathcal{L}_i(\theta)\| \leq L$ for all $\theta$
2. Both losses are locally convex in a neighborhood of convergence
3. Learning rate satisfies $\eta < \min\{1/L, 2/(\mu + L)\}$
4. Initial conflict exists: $\phi(\theta_0) < 0$

Then the expected gradient alignment converges to zero: $\lim_{t \to \infty} \mathbb{E}[\langle \mathbf{g}_1(\theta_t), \mathbf{g}_2(\theta_t) \rangle] = 0$.

#### A.2.3   Proof of Theorem A.1

**Step 1: Evolution of Individual Losses**

Using Taylor expansion around $\theta_t$, the expected change in $\mathcal{L}_1$ is:

$$\begin{aligned} \mathbb{E}[\mathcal{L}_1(\theta_{t+1})] &= \mathbb{E}\left[\mathcal{L}_1\left(\theta_t - \eta \mathbf{g}_t\right)\right] \\ &= \mathcal{L}_1(\theta_t) - \eta \mathbb{E}[\langle \mathbf{g}_1(\theta_t), \mathbf{g}_t \rangle] + \frac{\eta^2}{2} \mathbb{E}[\mathbf{g}_t^\top \mathbf{H}_1(\xi_t) \mathbf{g}_t] \end{aligned} \tag{7}$$

where $\mathbf{H}_1(\xi_t) = \nabla^2 \mathcal{L}_1(\xi_t)$ for some $\xi_t$ between $\theta_t$ and $\theta_{t+1}$.

Computing the expectation:

$$\mathbb{E}[\langle \mathbf{g}_1(\theta_t), \mathbf{g}_t \rangle] = (1 - p)\|\mathbf{g}_1(\theta_t)\|^2 + p\langle \mathbf{g}_1(\theta_t), \mathbf{g}_2(\theta_t) \rangle \tag{8}$$

Similarly for $\mathcal{L}_2$:

$$\mathbb{E}[\langle \mathbf{g}_2(\theta_t), \mathbf{g}_t \rangle] = p\|\mathbf{g}_2(\theta_t)\|^2 + (1 - p)\langle \mathbf{g}_1(\theta_t), \mathbf{g}_2(\theta_t) \rangle \tag{9}$$

**Step 2: Lyapunov Function Analysis**

Define the Lyapunov function:

$$V(\theta) = (1-p)\mathcal{L}_1(\theta) + p\mathcal{L}_2(\theta) + \alpha|\langle \mathbf{g}_1(\theta), \mathbf{g}_2(\theta)\rangle| \tag{10}$$

where $\alpha > 0$ is chosen small enough to ensure $V$ decreases.

The expected change in $V$ is:

$$\mathbb{E}[\Delta V] = -\eta\left[(1-p)^2\|\mathbf{g}_1\|^2 + p^2\|\mathbf{g}_2\|^2 + 2p(1-p)\langle \mathbf{g}_1, \mathbf{g}_2\rangle\right]$$
$$+ \alpha\mathbb{E}[\Delta|\langle \mathbf{g}_1, \mathbf{g}_2\rangle|] + O(\eta^2) \tag{11}$$

**Step 3: Dynamics of Gradient Alignment**

Let $A_t = \langle \mathbf{g}_1(\theta_t), \mathbf{g}_2(\theta_t)\rangle$. Using the chain rule:

$$\frac{dA_t}{dt} = \langle\nabla_\theta \mathbf{g}_1(\theta_t), \dot{\theta}_t\rangle^\top \mathbf{g}_2(\theta_t) + \mathbf{g}_1(\theta_t)^\top\langle\nabla_\theta \mathbf{g}_2(\theta_t), \dot{\theta}_t\rangle \tag{12}$$

Substituting $\dot{\theta}_t = -\eta\mathbb{E}[\mathbf{g}_t] = -\eta[(1-p)\mathbf{g}_1 + p\mathbf{g}_2]$:

$$\frac{dA_t}{dt} = -\eta\left[\mathbf{g}_{\mathrm{avg}}^\top\mathbf{H}_1\mathbf{g}_2 + \mathbf{g}_1^\top\mathbf{H}_2\mathbf{g}_{\mathrm{avg}}\right] \tag{13}$$

where $\mathbf{g}_{\mathrm{avg}} = (1-p)\mathbf{g}_1 + p\mathbf{g}_2$.

**Step 4: Linearization Near Critical Points**

Near a critical point where $\mathbf{g}_1$ and $\mathbf{g}_2$ are small, we can linearize. Let $\mathbf{g}_i = \mathbf{H}_i(\theta^*)(\theta - \theta^*)$ where $\theta^*$ is a local minimum of the combined loss. The alignment becomes:

$$A_t \approx (\theta - \theta^*)^\top\mathbf{H}_1\mathbf{H}_2(\theta - \theta^*) \tag{14}$$

**Step 5: Eigenvalue Analysis**

Let $\lambda_i^{(1)}$ and $\lambda_j^{(2)}$ be eigenvalues of $\mathbf{H}_1$ and $\mathbf{H}_2$ respectively, with corresponding eigenvectors $\mathbf{v}_i^{(1)}$ and $\mathbf{v}_j^{(2)}$. The contribution to alignment from the $(i, j)$ eigenspace pair is:

$$A_{ij} = \lambda_i^{(1)}\lambda_j^{(2)}|\langle\mathbf{v}_i^{(1)}, \mathbf{v}_j^{(2)}\rangle|^2 \cdot c_i c_j \tag{15}$$

where $c_i, c_j$ are the components of $(\theta - \theta^*)$ in the respective eigendirections.

**Step 6: Decay Rate**

The dynamics of each component follows:

$$\frac{dc_i}{dt} = -\eta\lambda_i^{\mathrm{eff}}c_i \tag{16}$$

where $\lambda_i^{\mathrm{eff}} = (1-p)\lambda_i^{(1)} + p\lambda_i^{(2)}\cos^2(\angle(\mathbf{v}_i^{(1)}, \mathbf{v}_i^{(2)}))$.

This gives exponential decay:

$$A_{ij}(t) = A_{ij}(0)\exp\left(-\eta(\lambda_i^{\mathrm{eff}} + \lambda_j^{\mathrm{eff}})t\right) \tag{17}$$

**Step 7: Convergence to Orthogonality**

Since all $\lambda_i^{\mathrm{eff}} > 0$ (by strong convexity), we have:

$$|A_t| \leq \sum_{i,j}|A_{ij}(t)| \leq C\exp(-\eta\lambda_{\min}t) \tag{18}$$

where $\lambda_{\min} = \min_{i,j}\{\lambda_i^{\mathrm{eff}} + \lambda_j^{\mathrm{eff}}\} > 0$ and $C$ is a constant depending on initial conditions.

Therefore:

$$\lim_{t\to\infty}A_t = \lim_{t\to\infty}\langle\mathbf{g}_1(\theta_t), \mathbf{g}_2(\theta_t)\rangle = 0 \tag{19}$$

Table 5: Result on human preference study in Likert scale of 1-5.

| Method | ID Consistency | Prompt Alignment | Motion Quality | Overall Quality |
|---|---|---|---|---|
| Omini+I2V | 3.80 | 3.78 | 3.62 | 3.44 |
| VideoBooth | 3.25 | 3.20 | 3.08 | 2.91 |
| Vidu 2.0 | 3.42 | 3.24 | 3.22 | 3.03 |
| **Ours** | **4.08** | **3.82** | **3.88** | **3.71** |

Table 6: **Ablation on the reference token.** Adding `<CLS>` yields improved subject identity scores (CLIP-I, DINO-I) and a higher dynamic degree.

| Training Method | Motion Smoothness | Dynamic Degree | CLIP-T | CLIP-I | DINO-I |
|---|---|---|---|---|---|
| w/o Ref. token | **98.84** | 54.55 | **32.87** | 73.36 | 57.51 |
| w/ Ref. token | 98.72 | **60.19** | 32.24 | **73.70** | **59.29** |

### A.2.4 COMPARISON WITH PCGRAD

PCGrad Yu et al. (2020) explicitly projects conflicting gradients:

$$\mathbf{g}_1^{\mathrm{PC}} = \begin{cases} \mathbf{g}_1 - \frac{\langle \mathbf{g}_1, \mathbf{g}_2 \rangle}{\|\mathbf{g}_2\|^2} \mathbf{g}_2 & \text{if } \langle \mathbf{g}_1, \mathbf{g}_2 \rangle < 0 \\ \mathbf{g}_1 & \text{otherwise} \end{cases} \tag{20}$$

Our method achieves $\langle \mathbf{g}_1, \mathbf{g}_2 \rangle \to 0$ through dynamics, effectively reaching the same orthogonal configuration without explicit projection. The convergence rate is:

$$\text{Ours: } O(e^{-\eta \lambda_{\min} t}) \quad \text{vs} \quad \text{PCGrad: } O(1) \text{ (immediate projection)} \tag{21}$$

While PCGrad achieves immediate orthogonalization, our method's gradual convergence allows for smoother optimization trajectories and better exploration of the loss landscape before settling into orthogonal configurations.

### A.3 HUMAN PREFERENCE STUDY

While benchmark metrics offer quantitative insights, they can sometimes be misled by "cheating" behaviors such as static outputs with artificially high scores. To complement our objective measurements, we conducted a human preference study using 20 randomly chosen samples from each baseline and our approach, *without cherry-picking*. A total of 30 participants were asked to rate the generated videos on a five-point Likert scale across dimensions of ID consistency, Prompt alignment, Motion quality, and Overall visual appeal. Our method consistently outperformed the baselines, suggesting that our balanced approach to identity preservation and temporal awareness best aligns with human judgments of video realism and quality when viewed holistically.

### A.4 ABLATION STUDIES

### A.4.1 EFFECT OF THE REFERENCE TOKEN

Tab. 6 demonstrates how adding a dedicated `<CLS>` token to the prompt affects our model's performance. Without this reference token, the model attains slightly higher motion smoothness (98.84) and marginally better CLIP-T (32.87), but it underperforms in dynamic degree (54.55) and identity-focused metrics (CLIP-I: 73.36, DINO-I: 57.51). Introducing `<CLS>` evidently improves subject fidelity (CLIP-I increases to 73.70 and DINO-I to 59.29) and fosters more diverse motion (dynamic degree rises to 60.19). We attribute these gains to the reference token guiding the alignment of subject tokens ($\mathbf{X}_{\mathrm{in}}$) with the textual prompt more explicitly, resulting in both stronger identity preservation and more coherent variations in motion.

Table 7: **Comparison with T2V-only stochastically-switched finetuning, with image drop probability of 1.** Switching to text-only input (T2V) moderately boosts CLIP-T but hurts subject fidelity and dynamic degree.

| Training Method | Motion Smoothness | Dynamic Degree | CLIP-T | CLIP-I | DINO-I |
|---|---|---|---|---|---|
| T2I + T2V (joint) | 98.85 | 44.14 | **33.41** | 72.71 | 48.68 |
| Ours | **98.72** | **60.19** | 32.24 | **73.70** | **59.29** |

Table 8: Abalation study on using different number of videos.

| Video Count | Motion Smoothness | Dynamic Degree | CLIP-T | CLIP-I | DINO-I |
|---|---|---|---|---|---|
| 1K | 99.03 | 59.66 | 32.16 | 72.69 | 56.98 |
| 2K | 98.96 | 52.25 | 32.49 | 72.13 | 54.42 |
| 3K | 98.79 | 55.46 | 32.04 | 72.79 | 55.57 |
| 4K (Ours) | 98.72 | 60.19 | 32.24 | 73.70 | 59.29 |

### A.4.2    T2V vs. I2V Training

We also ablate replacing our *image-to-video* (I2V) training with a text-to-video (T2V) setup. In Tab. 8, the T2I+T2V (joint) attains a slightly better CLIP-T but lower dynamic degree and subject alignment (CLIP-I, DINO-I). This suggests T2V training struggles when introducing a novel subject identity purely through text, yielding weaker overall identity preservation. By contrast, our I2V approach strikes a better balance, preserving subject details (CLIP-I: 73.70, DINO-I: 59.29) and maintaining sufficient motion (dynamic degree: 60.19).

### A.4.3    Effect of Varying the Video Dataset Size

Tab. 8 reports how our method's performance changes when using different amounts of unlabeled video data for I2V fine-tuning (1K, 2K, 3K, and 4K videos). Notably, with only 1K videos, we already obtain relatively strong results, suggesting that even a small unlabeled corpus can restore temporal consistency to some extent. However, increasing the video count to 4K (our default setting) steadily improves dynamic degree from 59.66 to 60.19 and also boosts identity fidelity (DINO-I) from 56.98 up to 59.29, indicating more consistent subject representation across frames.

We also observe a modest variation in CLIP-T and CLIP-I scores when moving from 1K to 4K videos, implying that a larger video dataset helps balance subject detail preservation and temporal motion, without overfitting to particular frames or motion patterns. In short, while our method is fairly robust to smaller unlabeled datasets, using around 4K (*i.e.*, 1% of Pexels jovianzm (2025) dataset) videos offers the best trade-off between data efficiency and stable motion/appearance results.

### A.4.4    Effect of Varying the Switching Probability

In Tab. 9, we examine how different values of $p$—the probability of sampling unlabeled video data (I2V fine-tuning)—affect overall performance. When $p = 0.0$, the model relies solely on image-based training (S2I) and achieves a relatively high dynamic degree (63.03) but moderate identity scores (CLIP-I: 72.86, DINO-I: 57.86). Increasing $p$ to 0.2 or 0.4 yields balanced improvements across most metrics, reflecting better coordination between identity and motion. At $p = 0.6$, the model dedicates a greater share of updates to I2V training, strengthening identity alignment (CLIP-I: 76.71, DINO-I: 62.84) while keeping dynamic degree stable (60.87). Although different $p$ values trade off between motion smoothness and identity fidelity to varying degrees, our chosen $p = 0.2$ demonstrates a strong overall balance, as highlighted in the main paper.

### A.5    Temporal Modeling Evaluation

We assess our model's capability to capture realistic object motion using a protocol adapted from FloVD Jin et al. (2025), while ensuring minimal or no camera movement in the test data.

Table 9: Abalation study on using different 'p' to choose I2V finetuning.

| Video Ratio | Motion Smoothness | Dynamic Degree | CLIP-T | CLIP-I | DINO-I |
|---|---|---|---|---|---|
| 0.0 | 99.60 | 0.84 | 32.67 | 71.15 | 43.19 |
| 0.2 (Ours) | 98.72 | 60.19 | 32.24 | 73.70 | 59.29 |
| 0.4 | 98.31 | 59.12 | 31.94 | 73.53 | 56.60 |
| 0.6 | 98.33 | 60.87 | 31.31 | 76.71 | 62.84 |

Specifically, we collect 100K videos from Pexels jovianzm (2025) that are *not* used during our stochastically-switched fine-tuning, then apply the following steps to create three benchmark subsets (*small*, *medium*, *large*) based on foreground motion magnitude:

**1) Foreground–Background Segmentation.** For each video, we use an off-the-shelf segmentation model (e.g., Grounded-SAM2) on the *first frame* to separate foreground and background regions. This allows us to measure object (foreground) motion independently from any camera-induced background shifts.

**2) Optical Flow Computation.** We estimate optical flow between the *first frame* and each subsequent frame using a standard flow estimator (e.g., RAFT Teed & Deng (2020)). Let $\mathbf{u}_f(x)$ and $\mathbf{u}_b(x)$ denote the per-pixel flow vectors for the foreground and background pixels, respectively, at position $x$. We record:

$$\text{FlowMag}_f = \frac{1}{N_f} \sum_{x \in \text{fg}} \|\mathbf{u}_f(x)\|,$$

$$\text{FlowMag}_b = \frac{1}{N_b} \sum_{x \in \text{bg}} \|\mathbf{u}_b(x)\|,$$

where $N_f$ and $N_b$ are the respective pixel counts in the foreground and background masks.

**3) Dataset Filtering.** To ensure negligible camera motion, we *discard* any video whose average magnitude of background flow $\text{FlowMag}_b$ exceeds 10 pixels. This filtering step excludes scenes with significant global shifts, retaining only those with primarily object-centric motion.

**4) Category Assignment.** Based on the average magnitude of foreground flow $\text{FlowMag}_f$ (averaged over all frames), we categorize videos into:

- *Small*: $0 \leq \text{FlowMag}_f \leq 25$
- *Medium*: $25 < \text{FlowMag}_f \leq 50$
- *Large*: $\text{FlowMag}_f > 50$

Each category contains 300 videos, ensuring a balanced evaluation of low-, moderate-, and high-motion scenarios.

**5) Evaluation Protocol.** Within each subset, we use only the *first frame* (including any textual or reference cues, if required) to generate a video of the same length. We then compute FVD Liu et al. (2024) between the generated outputs and the ground-truth videos. By comparing FVD across *small*, *medium*, and *large* motion classes, we obtain a clearer picture of how each model (ours vs. baselines) adapts to varying object-motion intensities.

**Discussion.** This motion-focused split highlights each method's strengths and weaknesses. For example, a model might produce near-static outputs for low-motion data—*cheating* on metrics like smoothness—yet fail to track fast-moving objects in high-motion videos. As observed in FloVD Jin et al. (2025), categorizing by foreground flow magnitude reveals these nuances more effectively than aggregated scores alone.

**Additional Details on Evaluation of Original CogVideoX.** To ensure a fair comparison with our method, we additionally fine-tune the original CogVideoX Yang et al. (2024) using a subset of the Pexels jovianzm (2025) dataset equivalent to the one used in our training. Since our model is trained for 4K steps with a sampling ratio $p = 0.2$, we match this by finetuning CogVideoX for 800 steps ($0.2 \times 4000$). As a result, we achieved a performance comparable to the original CogVideoX in terms of motion dynamics evaluation.

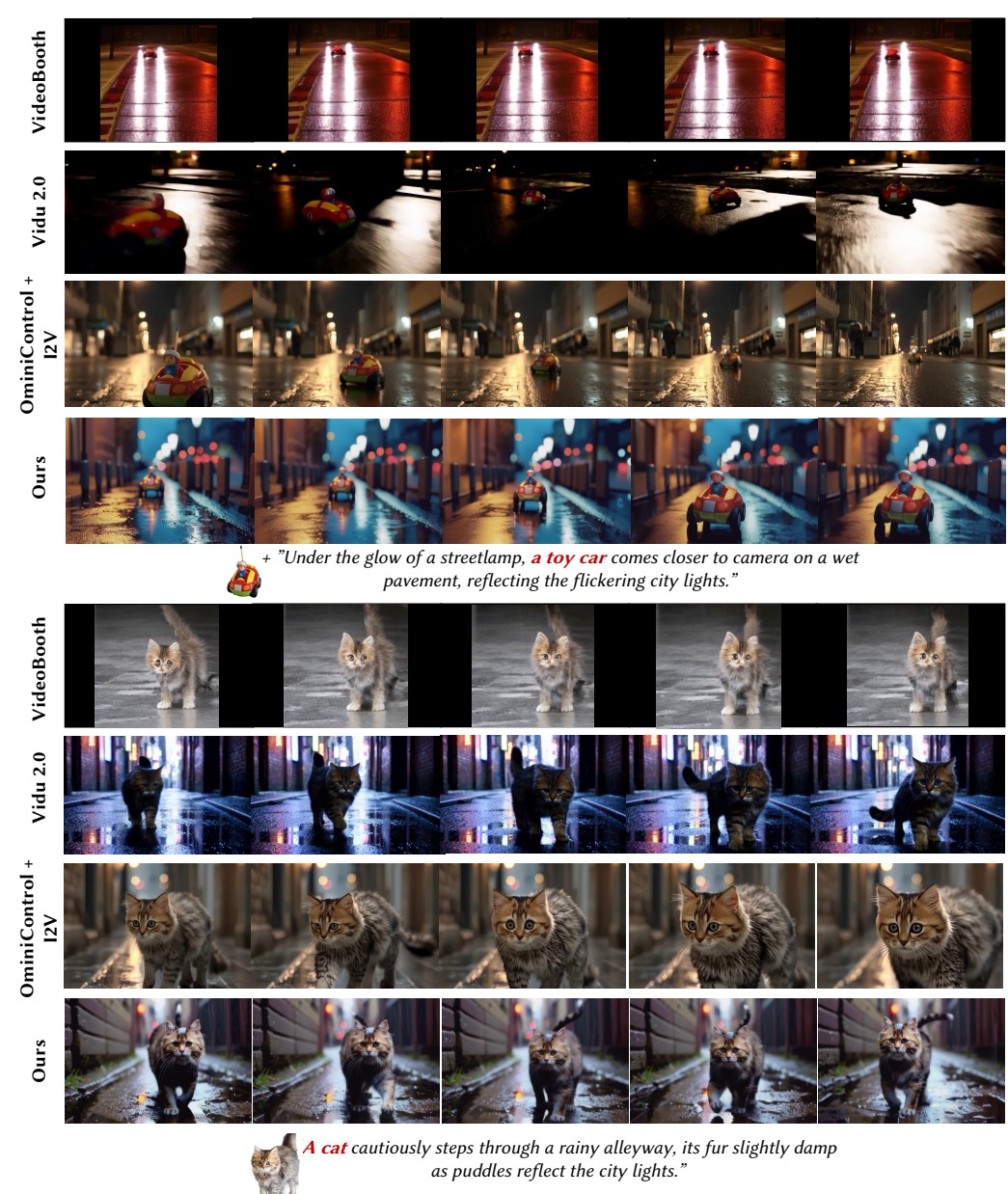

Figure 9: **Additional qualitative result.** Comparison with tuning-free baselines.

### A.6 QUALITATIVE RESULT WITH VIDEOS

We present additional qualitative results in Figure 9 and Figure 10, comparing with tuning-free baselines and tuning-based baselines. In Figure 9, we demonstrate that the OminiControl+I2V fails to interpret the 'comes closer to camera' for the first example with 'a toy car', due to the tendency not to generate close-ups as they tend to show articulated result as in Figure 4 in manuscript. Also in the example of 'Cat', they also show little articulated result with eye, since they fail to interpret small objects. Also compared to Vidu 2.0 and VideoBooth, they show degraded result compared to ours.

Additionally, when we compare with tuning-based baselines, ours show more identity-preserved result, compared to Still-Moving and CustomCrafter. For example, in the first example above with 'pig', ours follow the shape of 'pig' better, without change in colors or shapes of eyes. For the 'boy', we see that ours show less copy-pasted result, with heads turned during the video. For the

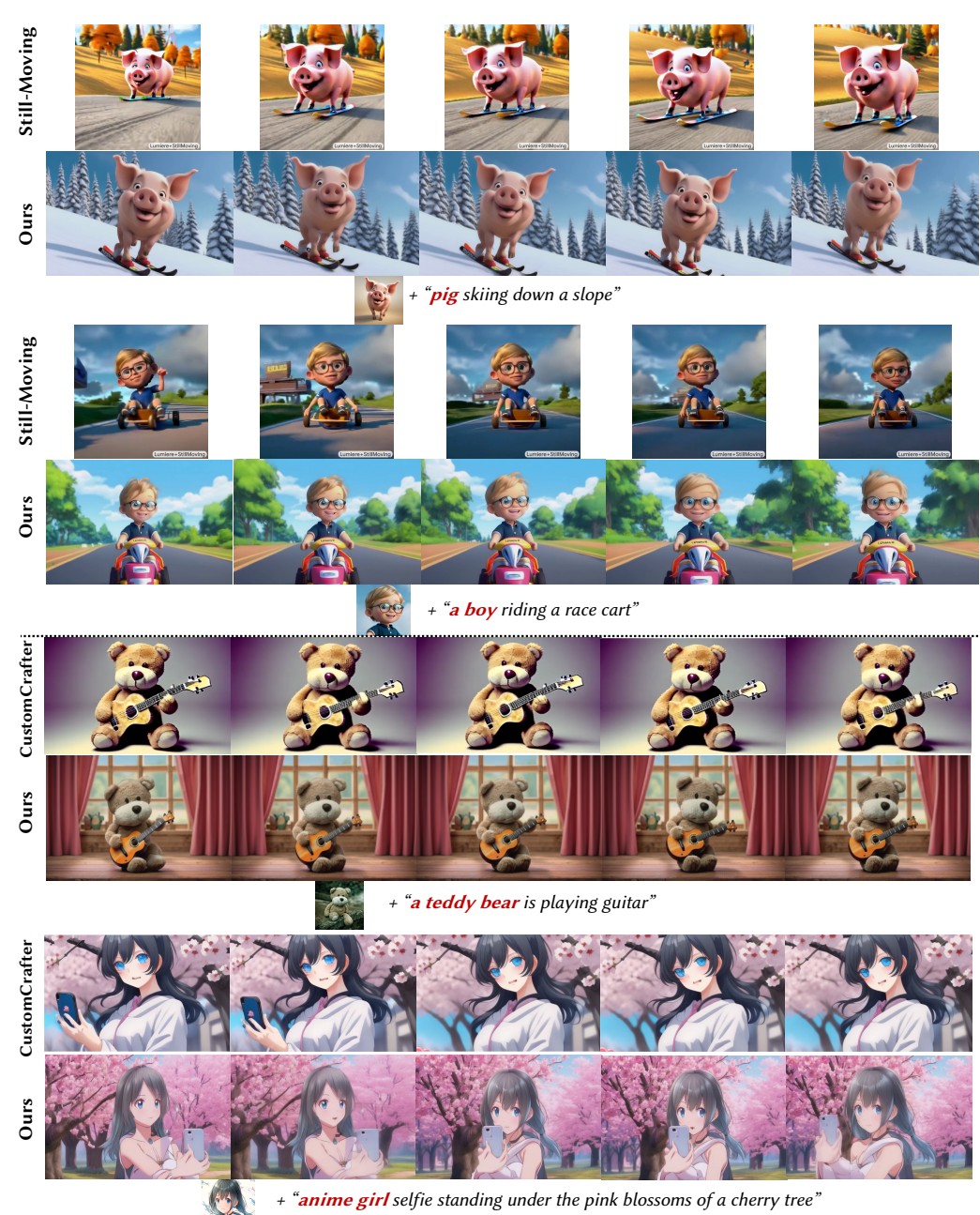

Figure 10: **Additional qualitative result.** Comparison with tuning-based baselines.

comparison with CustomCrafter in the third and fourth row examples, teddy bear's consistency also is better, also for the 'anime girl'. Please refer to the supplement video attached for moving videos.

## A.7 ESTIMATION OF OTHER METHODS' COMPUTATIONAL COST

We estimate the train time of CustomCrafter Wu et al. (2025) and VACE Jiang et al. (2025) based on the given implementation details on the manuscript.

**CustomCrafter.** The estimated GPU hours for training a single subject is in the range of 100–300 A100-hours, with a median estimate of around 200 A100-hours.

$$\text{wall-clock time (hours)} = \frac{10\,000 \times t}{3\,600}$$

$$\text{total GPU-hours} = 4 \times \text{wall-clock time}$$

where $t \in [10, 30]$ is the estimated time per iteration in seconds.

**VACE.** The estimated GPU hours for training VACE on LTX-Video is in the range of 9,000–27,000 A100-hours, with a median estimate of around 18,000 A100-hours.

$$\text{number of training steps} = 200\,000$$

$$\text{wall-clock time per training (hours)} = \frac{200\,000 \times t}{3\,600}$$

$$\text{GPU-hours} = 16 \times \frac{200\,000 \times t}{3\,600} = \frac{3\,200\,000t}{3\,600} = \frac{8\,000t}{9}$$

where $t \in [10, 30]$ is the estimated time per iteration in seconds. The estimated GPU hours for training VACE on Wan-T2V is in the range of 70,000–210,000 A100-hours, with a median estimate of around 140,000 A100-hours.

$$\text{number of training steps} = 200\,000$$

$$\text{wall-clock time per training (hours)} = \frac{200\,000 \times t}{3\,600}$$

$$\text{GPU-hours} = 128 \times \frac{200\,000 \times t}{3\,600} = \frac{25\,600\,000t}{3\,600} = \frac{64\,000t}{9}$$

where $t \in [10, 30]$ is the estimated time per iteration in seconds.

## B USE OF LARGE LANGUAGE MODELS

Anthropic's Claude was used to polish the writing of this manuscript. All text generated by the tool has been reviewed and revised by the authors.

