# OpenReview forum: "Subject-driven Video Generation Emerges from Experience Replays"
_ICLR.cc/2026/Conference — ICLR 2026 Conference Withdrawn Submission_

### Official Review · Reviewer_sJdV · 2025-10-16

**Soundness:** 3
**Presentation:** 1
**Contribution:** 2
**Rating:** 4
**Confidence:** 4

**Summary:**

This paper introduces a novel and highly efficient method for zero-shot, subject-driven video generation. The core contribution is a training strategy termed "proxy experience replay," which is founded on the hypothesis that learning a subject's identity from images and learning temporal dynamics from videos are orthogonal objectives. The authors experimentally validate this hypothesis, demonstrating that the gradient conflict between these tasks naturally converges to near-zero.
Based on this theoretical analysis, the author conducted an experiment and reduced the training cost of subject to video by using subject to image data mixed with a small amount of video data for training.

**Strengths:**

1.  The paper's idea of using image data and a small amount of video data to accomplish Subject-to-Video generation is good.
2.  The method of analyzing the feasibility of using image data and a small amount of video data for the Subject-to-Video generation task from the perspective of gradients is very interesting.

**Weaknesses:**

1. Although the author's method of analyzing from the perspective of gradients is very interesting, I still have some questions:

    a. The paper only provides the cosine similarity of the gradients of the two tasks during training when using proxy replay, lacking corresponding control experiments, i.e., training models with the S2I task and the T2V task separately, and counting the gradients individually to see if there is a conflict.

    b. The 'gradient conflict' metric, defined as the cosine similarity $\phi(t)$ in Eq. 2, is theoretically invariant to the scale of the gradients. However, this property may not hold in practice during the later stages of training. As the model converges, the magnitudes (L2 norms) of the gradients, $\|\nabla L_{\text{img}}\|$ and $\|\nabla L_{\text{vid}}\|$, tend to decay towards zero. When these magnitudes become sufficiently small, the computation of $\phi(t)$ can become numerically unstable. The observed trend of $\phi(t) \to 0$ might therefore be an artifact of vanishing gradients, where near-zero vectors can appear spuriously orthogonal due to floating-point inaccuracies, rather than a genuine indicator of directional orthogonality. Specifically, the author should add the monitoring results of the gradient magnitudes. If $\phi(t)$ decreases while $\|\text{g}_1\|$ and $\|\text{g}_2\|$ remain stable, it indicates that the directional change is real.
2. The technical contribution is weak. Fundamentally, the method proposed in this paper is to train a video generation model using a mix of image and video data. However, this operation has been widely used in the training of various Text-to-Video foundation models, such as VideoCrafter2[1], Open-Sora[2], etc. Applying it to the S2V task is just a simple change of the training task.
3. Many of the compared methods are based on UNet, such as Videobooth, Stilling Moving, Customcrafter. The base models of these methods have a large gap with the CogVideoX-5b used by the author. There is a lack of comparison with some new methods based on DiT architecture with comparable base model performance, such as ConsisID[3], VACE[4], and Phantom[5]. The author could consider using the existing OpenS2V-Eval BenchMark[6] for comparison.
4. The generated results do not look very good visually. Are there results based on other foundation models, for example, Wan2.2?

5. Minor Suggestions
* Equations (2) and (3) are missing closing punctuation.
* Please use the `\citep` command from the ICLR template correctly for citations. The citation format in the paper is severely disorganized.

If the author addresses my concerns and revises the article, I will increase my score.

[1] Videocrafter2: Overcoming data limitations for high-quality video diffusion models

[2] Open-Sora Plan: Open-Source Large Video Generation Model

[3] Identity-Preserving Text-to-Video Generation by Frequency Decomposition

[4] VACE: All-in-One Video Creation and Editing

[5] Phantom: Subject-Consistent Video Generation via Cross-Modal Alignment

[6] OpenS2V-Nexus: A Detailed Benchmark and Million-Scale Dataset for Subject-to-Video Generation

**Questions:**

See Weakness

---

### Official Review · Reviewer_wVzP · 2025-10-30

**Soundness:** 3
**Presentation:** 3
**Contribution:** 3
**Rating:** 6
**Confidence:** 4

**Summary:**

This paper presents a method for subject-driven video generation, focusing on maintaining identity consistency across frames. The proposed approach introduces a cross-frame synchronization mechanism that aligns subject features over time using both temporal feature linking and latent-space regularization. The model builds upon a pretrained video diffusion transformer, incorporating reference-frame encoding and temporal consistency loss to stabilize identity appearance. The authors also propose a SubjectSync benchmark for quantitative evaluation, covering identity preservation, motion realism, and text alignment.

**Strengths:**

* The paper addresses a persistent challenge in personalized video generation: temporal drift in subject identity. The motivation and proposed synchronization perspective are both well-articulated.

* The cross-frame synchronization module is conceptually clean and lightweight. Its integration with existing diffusion transformers is practical and easy to reproduce.

* SubjectSync provides a useful resource for measuring temporal and identity consistency in personalized video generation.

**Weaknesses:**

* The main technical idea (temporal feature linking with feature-level regularization) is a straightforward extension of existing feature-consistency approaches used in image personalization and animation diffusion.

* The proposed synchronization loss depends on reference frames and precomputed feature correspondences, which may limit applicability to videos with large pose or viewpoint changes.

* The experiments primarily involve human-centric data. It remains unclear whether the method generalizes to non-human or stylized subjects.

* The ablations focus on module removal but do not provide insights into design sensitivity, e.g., synchronization weight λ, number of reference frames, or frequency of linking.

**Questions:**

Most of the major concerns are reflected in the weaknesses above, while I still have several minor questions as follows.

* How robust is the synchronization mechanism to appearance drift caused by lighting or camera motion, especially when the subject’s features are partially occluded?

* Could the cross-frame synchronization mechanism be adapted for multi-subject videos, or does it rely on a single dominant subject embedding?

---

### Official Review · Reviewer_iWod · 2025-11-01

**Soundness:** 3
**Presentation:** 3
**Contribution:** 2
**Rating:** 4
**Confidence:** 5

**Summary:**

This paper targets efficient subject-to-video (S2V) learning without relying on costly video–subject-pair datasets. The authors observe that naively training video models with image-paired data induces catastrophic degradation of temporal coherence due to gradient conflicts. They hypothesize that S2V can be decomposed into two orthogonal objectives: identity learning from images and temporal dynamics from videos. Building on this, they propose a stochastic task-switching strategy that predominantly samples from image datasets while maintaining minimal video replay to preserve temporal ability. Empirically, they show that the gradient inner product between the two tasks converges rapidly toward zero, indicating emergent orthogonalization without explicit projection. Experiments demonstrate superior performance with compute comparable to per-subject tuned methods for single-subject customization, while also providing zero-shot capability and outperforming both per-subject tuned approaches and several existing zero-shot baselines.

**Strengths:**

1. The manuscript is clearly written and easy to follow.
2. The motivation is well articulated: addressing the heavy reliance on expensive video data for zero-shot video customization.
3. The topic is valuable, aligning with personalized user needs and real-world applications.
4. The paper presents extensive experiments and some insightful empirical analyses.

**Weaknesses:**

1. The decomposition of video customization into ID injection and temporal dynamics appears similar to prior work (e.g., DreamVideo, Still-Moving). Please clarify the conceptual and methodological differences from these approaches, and specify what is genuinely new.
2. The connection to continual learning (CL) is unclear. In standard CL, models learn sequentially over tasks/data while mitigating catastrophic forgetting and acquiring new capabilities. Here, the missing temporal dynamics seem to stem from fine-tuning on static image data rather than from sequential learning.
3. The methodological novelty seems limited: the ID injection component does not introduce a new design, and the proposed “video replay” looks like joint training on images and videos rather than a CL-style memory replay mechanism, as the replay buffer is not updated during training.
4. The tuning-free baselines used for comparison appear dated. Stronger and more recent video customization baselines would make the evaluation more convincing.
5. Does the proposed method support multi-subject customization?

**Questions:**

If my concerns are satisfactorily addressed, I would be happy to raise my score.

---

### Official Review · Reviewer_scwL · 2025-11-09

**Soundness:** 3
**Presentation:** 3
**Contribution:** 2
**Rating:** 4
**Confidence:** 4

**Summary:**

This paper tackles the problem of subject-driven video generation (S2V) without relying on expensive subject–video paired datasets. The authors propose to decompose S2V learning into two independent objectives: learning subject identity from images and learning temporal dynamics from generic unpaired videos. Based on this idea, they introduce a simple stochastic task-switching strategy that alternates between image and video batches during training, allowing the model to learn identity consistency mainly from images while capturing motion coherence from a small set of unpaired videos. Experiments demonstrate that the proposed method enables zero-shot subject-to-video generation with strong temporal and identity consistency, achieving competitive or superior visual quality to existing methods while requiring much lower computational cost.

**Strengths:**

1. The method generates temporally coherent and identity-consistent videos with lower computational cost than baselines.
2. The method enables subject-driven video generation without costly paired datasets, addressing a real limitation in current approaches.
3. The paper is well written, logically structured, and supported by clear figures and examples.

**Weaknesses:**

1. The core idea of alternating between image and video training is conceptually similar to multi-task training strategies, suggesting limited methodological novelty.
2. The paper acknowledges "blurry artifacts" in human face generation, and the paper attributes this to a lack of face samples in the dataset. However, no further experiments are provided to confirm whether the issue arises from data imbalance or from the method’s inability to model fine-grained facial features.
3. The paper does not include comparisons with recently published zero-shot S2V works, which would undermine confidence in the conclusions.
4. Can your method be readily applied to video editing tasks?  If so, could you conduct some experiments to validate its performance?

**Questions:**

The evaluation appears to be conducted exclusively on short video clips. How does the method perform on longer sequences? How does the temporal consistency of this method compare to the baselines over time?

---

### Note · Authors · 2025-11-14

**Comment:**

We thank the reviewers for their constructive feedback and encouraging comments on our approach. We appreciate recognition of the framework's strengths and feedback regarding weaknesses. Comments would be invaluable in helping us refine and strengthen our research approach. After careful consideration, We have decided to withdraw our paper from this year’s review process to allow for further development.

**Withdrawal Confirmation:**

I have read and agree with the venue's withdrawal policy on behalf of myself and my co-authors.